

# Enhanced object detection of *Enterobius vermicularis* eggs using cumulative transfer learning algorithm

Pongphan Pongpanitanont[1], Naparat Suttidate[1,2,3], Hiroshi Yamasaki[4], Wanchai Maleewong[5,6] and Penchom Janwan[1,7,8]

[1] Health Sciences (International Program), College of Graduate Studies, Walailak University, Nakhon Si Thammarat, Thailand
[2] Akkhraratchakumari Veterinary College, Walailak University, Nakhon Si Thammarat, Thailand
[3] Centre for One Health, Walailak University, Nakhon Si Thammarat, Thailand
[4] Department of Parasitology, National Institute of Infectious Diseases, Tokyo, Japan
[5] Department of Parasitology, Faculty of Medicine, Khon Kaen University, Khon Kaen, Thailand
[6] Mekong Health Science Research Institute, Khon Kaen University, Khon Kaen, Thailand
[7] Department of Medical Technology, School of Allied Health Sciences, Walailak University, Nakhon Si Thammarat, Thailand
[8] Hematology and Transfusion Science Research Center, Walailak University, Nakhon Si Thammarat, Thailand

Corresponding author
Penchom Janwan,
pair.wu@gmail.com

## ABSTRACT

Traditional diagnostic methods in medical parasitology rely heavily on manual microscopic examination, which is labor-intensive and prone to human error and subjectivity. This study introduced a novel approach for automating the detection of *Enterobius vermicularis* (pinworm) eggs using cumulative transfer learning algorithms. The proposed framework effectively captures subtle egg morphology by employing a sequential knowledge transfer paradigm, thereby enhancing diagnostic accuracy, efficiency, and reproducibility, even when data are limited. This study used *E. vermicularis* egg images from a publicly available dataset. The training image dataset comprised 1,000 images of artifacts and 1,000 images of pinworm eggs. Comparisons were made against established deep learning (DL) models, including conventional convolutional neural network (CNN), ResNet50, DenseNet121, Xception, and InceptionV3. Results demonstrated that the cumulative transfer learning strategy consistently outperformed both the conventional CNN method and DL baselines in terms of classification accuracy, F1-score, and computational efficiency, while also reducing computational overhead. Performance comparison with a conventional CNN model demonstrates that the proposed cumulative transfer learning CNN reduces training time from 2 h to 50 min. Moreover, it achieves optimal performance, with accuracy, precision, recall, and F1-score all reaching 1.0. The model's detection accuracy was quantitatively assessed by comparing predicted bounding boxes to expert annotations across 103 microscopic images. The proposed cumulative transfer learning CNN achieved higher average precision (AP) @ intersection over union (IoU) 0.5 (0.530) and perfect sensitivity (1.00), but exhibited 97 false positives and lower mean average precision (mAP) @IoU0.5:0.05:0.95 (0.027). In contrast, the You Only Look Once version 8 (YOLOv8) model demonstrated lower sensitivity (0.72) but superior multi-threshold performance (mAP@IoU0.5:0.05:0.95 = 0.057). These results highlight a trade-off between detection sensitivity and generalization performance across varying IoU thresholds. These findings affirm the viability of cumulative transfer learning as a scalable,

accurate, and efficient approach for automated parasitological diagnostics, particularly in resource-limited settings.

## INTRODUCTION

*Enterobius vermicularis* (pinworm) represents a ubiquitous nematode parasite with significant global prevalence, demonstrating particular pediatric populations in the tropics *via* fecal-oral transmission and ova inhalation. Although clinical manifestations are frequently subclinical, characteristic perianal pruritis predominates, with potential progression to appendicular and genitourinary complications (*Taghipour et al., 2020*; *Lashaki et al., 2023*; *Naqvi, Atarere & Parungao, 2023*). Emerging evidence also indicates that *E. vermicularis* infection can impact gut microbiota, increasing its diversity while lowering intestinal secretory immunoglobulin A (IgA) levels, thereby influencing immune responses (*Yang et al., 2017*). The global epidemiology of enterobiasis (pinworm infection) demonstrates marked geographical heterogeneity, with a pooled prevalence of 12.9% among pediatric populations. Regional surveillance studies reveal substantial variation across continents: Asian prevalence rates range from 4.4% in the Republic of Korea to 55% in China, with intermediate rates reported in Thailand (8.8%), Kyrgyzstan (19.3%), and Myanmar (47.2%). African studies document prevalences of 26.3%, 11.7%, and 1.7% in Tanzania, Nigeria, and Angola, respectively. In South America, Chile reports 35% prevalence, while Argentina shows 19%. European data, as exemplified by Germany, indicates a prevalence of 17.4% (*Lashaki et al., 2023*). Socioeconomic and environmental factors, including sanitation and personal hygiene, remain the key determinants of pinworm transmission dynamics in different regions (*Wang, Hwang & Chen, 2010*). Management of *E. vermicularis* infestation encompasses anthelmintic pharmacotherapy, predominantly utilizing mebendazole, pyrantel pamoate, or albendazole (*CDC, 2024*). To ensure optimal treatment efficacy, laboratory-confirmed diagnosis through microscopic examination is essential before therapeutic intervention. The high transmissibility of enterobiasis in communal environments necessitates expeditious and precise diagnostic protocols to interrupt transmission cycles and effectively minimize population-level health implications. The gold standard diagnostic strategy involves perianal swabs, with the scotch tape technique recognized for its simplicity, cost-effectiveness, and high diagnostic yield (*Wendt et al., 2019*).

Contemporary molecular diagnostic methodologies, specifically polymerase chain reaction, demonstrate enhanced analytical sensitivity and specificity. However, their widespread implementation is constrained by substantial financial requirements and specialized infrastructure needs, particularly in resource-limited settings. The scotch tape technique maintains its practical significance and diagnostic efficacy, especially in

large-scale surveillance programs within low-resource environments (*Bharadwaj et al., 2021*; *Deroco, Junior & Kubota, 2021*; *Jayakody et al., 2022*). Nevertheless, this approach exhibits inherent limitations. Diagnostic precision depends on multiple variables including microscopist expertise, workload, non-standardized practices, and human error from exhaustion. To mitigate these constraints, integrating traditional scotch tape technique with advanced computer vision technologies presents a synergistic diagnostic paradigm. This approach, particularly through artificial intelligence (AI)-driven analytical models, potentially enhances diagnostic accuracy and facilitates result validation in high-throughput clinical microscopy services and resource-constrained settings. Recent advances in computer vision technologies have demonstrated promising applications in parasitological diagnosis. Several studies have explored machine learning (ML) approaches for parasite detection and classification. *Liang et al. (2016)* demonstrated a novel convolutional neural network (CNN) model designed to automatically classify malaria-infected red blood cells in blood smear images, achieving an impressive accuracy of 97.4%. *Holmström et al. (2017)* compared the results of digital image analysis using a mobile microscope and commercially available image analysis software WebMicroscope-deep learning (DL)-based algorithms software to the manual labeling of soil-transmitted helminth (STH) eggs in the images by the researchers. The detection sensitivity of *Ascaris lumbricoides* was 100%, that of *Trichuris trichiura* was 83.3%, and that of hookworm eggs was 93.8%. *Lundin et al. (2024)* demonstrated that deep learning systems (DLS) analyzing whole-slide images (WSIs) of Kato-Katz thick smears achieved robust diagnostic performance for STHs, with a sensitivity of 76.4–91.9% and specificity of 89.7–98.2% compared to manual microscopy. Notably, DLS-enabled digital microscopy detected light-intensity infections that were overlooked by conventional manual examination. *Thanchomnang et al. (2024)* proposed a CNN for classifying and automatically detecting *Opisthorchis viverrini*, carcinogenic liver fluke eggs from digitized images. The model trained with augmented data remarkably reached the pinnacle of accuracy, scoring 1.00. An average intersection over a union (IoU) score exceeding 0.5 yielded 69.47%. In a recent study, *Chaibutr et al. (2024)* employed a conventional CNN architecture for automated *E. vermicularis* egg detection in microscopic images, achieving 90.0% accuracy. However, advanced architectures such as Xception and Resnet50 models demonstrated superior performance, achieving 99.0% accuracy.

Transfer learning is a methodological framework where a model leverages prior knowledge from one domain (source task) to improve performance on another, typically data-scarce, domain (target task) (*Pan, 2013*; *Hosna et al., 2022*). This technique eliminates the need for extensive labeled data and accelerates training on new tasks. It offers marked advantages over traditional ML methods that rely heavily on large, domain-specific datasets. Cumulative transfer learning extends this paradigm by sequentially transferring knowledge across multiple tasks. Through iterative refinement of inductive biases, models develop more robust internal representations and exhibit superior adaptability and generalization (*Székely et al., 2022*). This iterative knowledge transfer reduces training time and data requirements, making it particularly beneficial in specialized domains like fluid dynamics (*Inubushi & Goto, 2020*) and robotics, where systems must consistently adapt to

**Table 1 Summary of related studies on parasitic egg detection, highlighting each study's main findings, research gaps, and methodological limitations.**

| Study | Conclusion | Research gap | Limitations |
|---|---|---|---|
| *He et al. (2024)* | The You Only Look Once version 4 (YOLOv4) algorithm achieves high accuracy in detecting parasitic helminth eggs, including 100% for *Clonorchis sinensis* and *Schistosoma japonicum*. | The need for further optimization of the artificial intelligence model to enhance detection consistency, particularly for species with lower accuracy rates. | The model is effective but varies in accuracy across species and mixed infections, requiring further optimization. |
| *Xu et al. (2024)* | The YOLOv5n + AFPN + C2f network (YAC-Net) model improves detection performance over YOLOv5n while reducing complexity, enabling efficient parasite egg detection in resource-limited settings. | YAC-Net concerning its performance on diverse parasitic eggs, variations in microscopy image quality, and long-term adaptability in real-world applications requiring periodic updates and retraining. | YAC-Net's improvements but does not address its real-world limitations, such as image quality requirements and model generalizability across parasite species. |
| *Wan et al. (2023)* | Composite backbone network (C2BNet) improves parasitic egg detection with a two-path structure and multiscale weighted box fusion, enhancing feature learning and detection accuracy over existing methods. | An unexplored aspect of C2BNet's computational efficiency, scalability, and generalizability, which limits its applicability to diverse parasitic infections and broader microscopic imaging tasks. | A result shows imaging challenges but omits C2BNet's limitations. |
| *Penpong et al. (2023)* | The two-step approach improves out-of-domain parasite egg detection, boosting F1-score to 57.97% and enhancing performance with threshold strategies, achieving up to 77.30%. | The limited evaluation of diverse unrelated objects and the lack of exploration of alternative optimization methods for out-of-domain detection. | The out-of-domain (OO-Do) problem lowers accuracy, and despite the two-step SoftMax improving detection (F1-score: 57.97%), it remains below the 77.30% achieved without OO-Do. |
| *Pho et al. (2022)* | The attention-driven RetinaNet improves parasitic egg detection, refining segmentation with Guided- and self-attention, achieving a mean average precision (mAP) of 0.82 on the IEEE dataset. | The limited segmentation annotation and poor generalizability of attention-driven RetinaNet due to reliance on small datasets. | Limited segmentation annotations may reduce detection accuracy. |

new scenarios (*Jaquier et al., 2023*). However, pivotal challenges remain. Negative transfer can occur when source and target tasks are insufficiently related, causing degraded performance in the target domain. Additionally, determining precisely what knowledge to transfer remains a critical issue, as improperly aligned features can hinder rather than help performance (*Muller et al., 2019*). Nevertheless, continued advancements in transfer and a cumulative transfer learning strategies demonstrate their growing significance and considerable potential for driving innovative solutions in various ML applications (*Kowald et al., 2022*). While DL has advanced automated parasite detection, current methods for identifying *E. vermicularis* eggs remain limited by require extensive data and computational resources (*Chaibutr et al., 2024*). Most studies have not investigated cumulative transfer learning, which could improve accuracy and adaptability in settings with limited or artifact-rich data (*Liang et al., 2016*; *Holmström et al., 2017*; *Pho et al., 2022*; *Penpong et al., 2023*; *Wan et al., 2023*; *Chaibutr et al., 2024*; *He et al., 2024*; *Lundin et al., 2024*; *Thanchomnang et al., 2024*; *Xu et al., 2024*). Thus, there remains a critical need for robust, efficient AI-based frameworks specifically tailored to the automated detection of *E. vermicularis* eggs in real-world, resource-constrained environments. This study proposes an innovative cumulative transfer learning algorithm that revolutionizes object

detection capabilities for *E. vermicularis* eggs, offering substantial improvements over conventional approaches.

## Related works

Table 1 provides a comprehensive comparison of related works with other studies, specifically analyzing their conclusions, identified research gaps, and methodological limitations. This comparison underscores the novelty of our research, which introduces an innovative approach for the detection of *E. vermicularis* eggs utilizing the scotch tape technique. By addressing the limitations of previous studies, our proposed method enhances diagnostic accuracy and efficiency, contributing to advancements in parasitological detection techniques.

# MATERIALS AND METHODS

## Data collection

This study used *E. vermicularis* egg images from a publicly available dataset accessible through Figshare (https://doi.org/10.6084/m9.figshare.26266028.v2) (*Chaibutr et al., 2024*). The *E. vermicularis* eggs were collected using the established scotch tape method. A piece of transparent adhesive tape was gently pressed against the perianal region of patients, typically first thing in the morning before bathing or defecation, to collect nocturnal egg deposits. The tape was then affixed onto a glass slide and examined microscopically to confirm the presence of characteristic *E. vermicularis* eggs. Confirmed eggs were photographed using a digital microscope camera for subsequent image processing and analysis. Each image was manually inspected and annotated by experienced laboratory personnel. The annotated images were separated into two distinct classes: (1) class 0 for negative samples or artifact structures with 1,000 images, and (2) class 1 for positive samples of 1,000 images exhibiting the characteristic morphology of *E. vermicularis* eggs (Figs. 1A, 1B). This manual curation process ensured that only high-quality, correctly labeled samples were included in the training and testing sets. The dataset comprised 2,000 high-resolution (2,448 × 1,920 pixels) Tagged Image File Format images with balanced distribution between *E. vermicularis* eggs and artifacts. For the training and validation, we utilized the Google Colab platform with robust computational resources: an NVIDIA A100 GPU, Intel Xeon CPU operating at 2.3 GHz, 83.5 GB of RAM, and 235.7 GB of disk space.

The study protocol was approved by the Ethics Committee in Human Research, Walailak University (protocol code WUEC-24-387-01, approved 8 November 2024) and followed the Declaration of Helsinki. The dataset was distributed under the Creative Commons Attribution (CC BY) license and contained only microscopic images of artifacts and *E. vermicularis* eggs, with no personal identifiable information or sensitive data.

## Dataset preparation

Generalizability and robustness of the trained model, the original dataset underwent several advanced image augmentation procedures. Gaussian blur was introduced to replicate variations in focus commonly observed in real-world microscopic imaging,

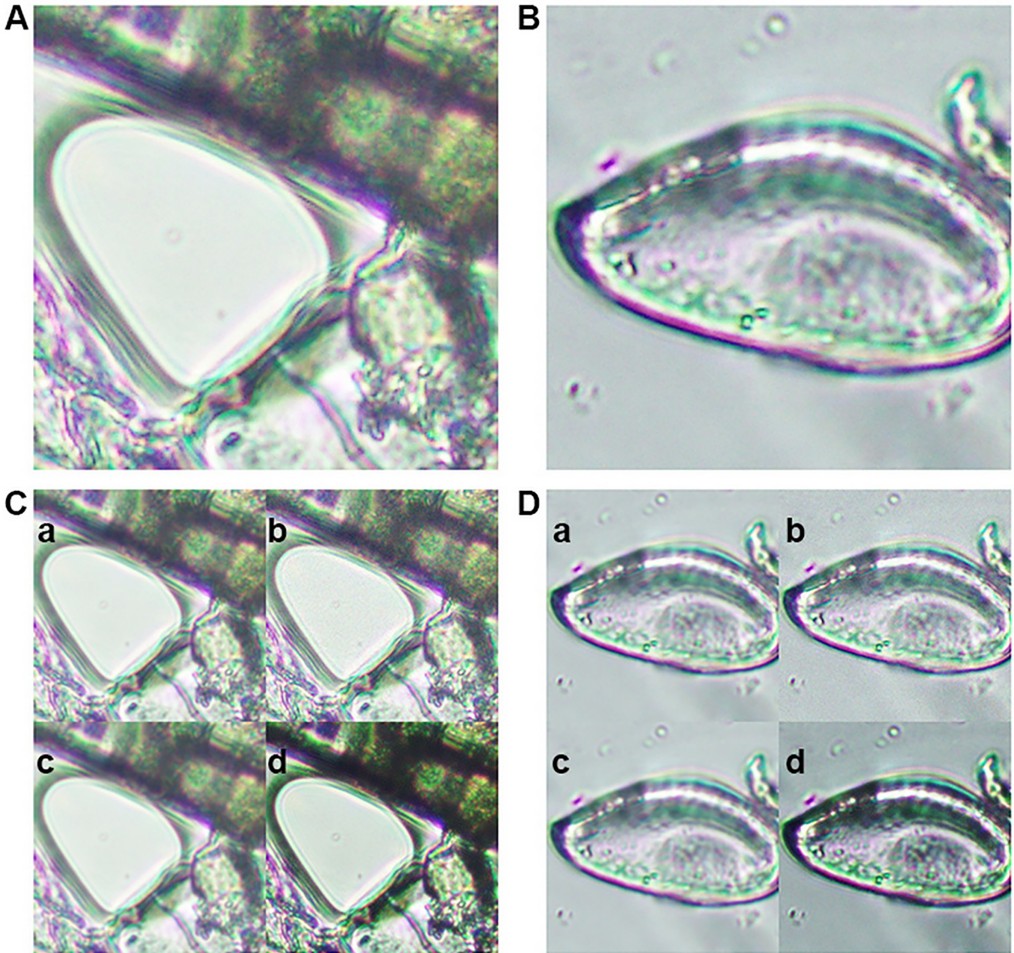

**Figure 1 Representative images from the training dataset.** (A) Microscopic image of an artifact (class 0). (B) Microscopic image of an *Enterobius vermicularis* (pinworm) egg (class 1). Examples of image augmentation techniques applied to (C) artifact images and (D) pinworm egg images: (a) Gaussian blur, (b) Gaussian noise addition, (c) mean filtering, and (d) sharpening.

thereby enabling the model to handle slightly out-of-focus samples. Mean filtering was employed to smooth images and mitigate high-frequency noise, while Gaussian noise addition mimicked the random intensity fluctuations frequently encountered in microscopic observations. Finally, kernel sharpening was applied to emphasize edges and subtle morphological details of the eggs, ensuring that essential diagnostic features remained discernible (Figs. 1C, 1D). By integrating these augmentation techniques, the dataset's diversity was substantially increased, resulting in improved model performance and enhanced resilience to image quality variations commonly encountered in clinical applications. This approach enables the model to maintain robust detection capabilities when processing unseen data with varying imaging conditions, such as differences in microscope settings, illumination levels, and sample preparation protocols. The enhanced dataset diversity ensures that the trained model can generalize effectively across different

laboratory environments and imaging equipment configurations (*Chaibutr et al., 2024*; *Lundin et al., 2024*; *Thanchomnang et al., 2024*).

## Model construction and learning

In this study, a five-fold cross-validation protocol was employed to rigorously evaluate model performance, ensuring that each subset of the dataset served as both training and validation data in the separate runs for a more reliable measure of generalizability. The network was trained for a maximum of 200 epochs unless the mean-squared-error (MSE) loss fell below 0.001, which acted as an early stopping criterion (*Tuite et al., 2011*). The model architecture comprised the three sequential blocks of convolutional layers of decreasing dimension in terms of filter count. Each block contained three consecutive Conv2D layers with ReLU activation initially 64 filters in the first block, then 32 filters in the second, and finally 8 filters in the third block, followed by a MaxPooling2D operation to reduce spatial dimensionality. The feature maps were then flattened and passed through four fully connected (Dense) layers, sized 256, 64, 16, and two neurons, respectively. Each layer employed the sigmoid activation function to progressively distill the essential features and ultimately output the classification results. The network was compiled using a stochastic gradient descent (SGD) as the optimizer and the MSE loss function, with accuracy tracked as the primary performance metric. This configuration aimed to balance computational efficiency with sufficient modeling capacity, leveraging progressive feature extraction in the convolutional layers and feature refinement in the dense layers for robust egg detection.

Conventional training (Fig. 2A) in a supervised learning context typically began with importing the dataset from a designated source, such as an image repository or a curated collection of labeled samples. Once imported, data augmentation is performed to artificially expand and diversify the dataset by applying transformations such as rotation, flipping, or slight color modifications, while preserving the essential features necessary for accurate classification or detection. This augmented dataset was then fed into a selected model architecture, where training was conducted to optimize the model's parameters through iterative updates driven by a chosen loss function and optimization algorithm. Finally, the validation phase was performed using a portion of the data that was held out, enabling the assessment of the model's performance and generalization capabilities. This conventional pipeline seeks to balance the model's ability to learn robust representations with the need to avoid overfitting, ultimately aiming to achieve high accuracy on unseen data (*Miseta, Fodor & Vathy-Fogarassy, 2024*).

Cumulative transfer learning (Fig. 2B) refines the conventional supervised learning pipeline encompassing data collection, labeling, augmentation, and model training by integrating a sequential, knowledge-preserving methodology (*Yang et al., 2019*). The process begins with the assembly of an annotated dataset, which is then augmented to enhance the breadth of feature patterns and mitigates overfitting. Rather than instantiating model weights from scratch, training commences by importing those from a previously optimized model, thereby exploiting established feature representations (*Liu et al., 2020*). Crucially, the incoming augmented dataset was divided into smaller subsets, each

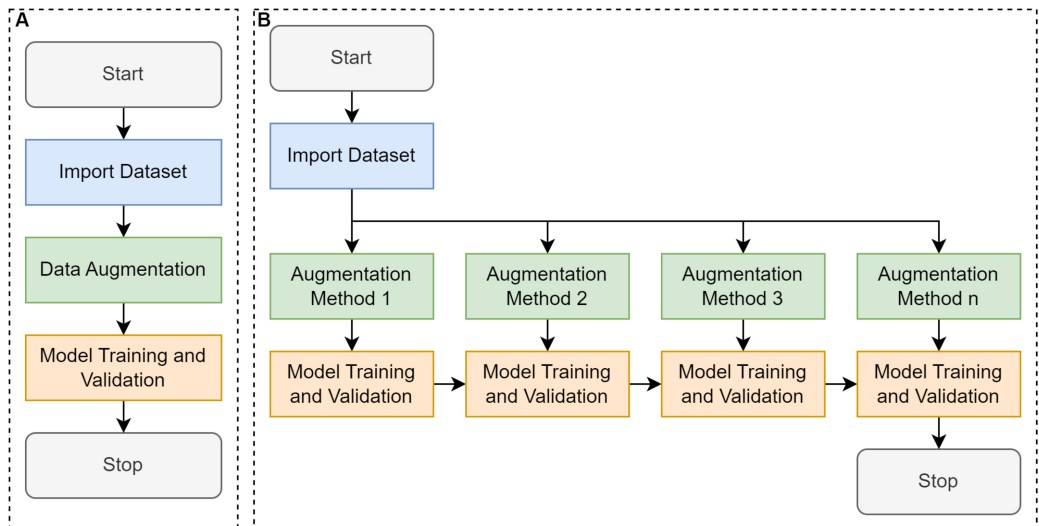

**Figure 2 Comparative workflow between conventional learning and cumulative transfer learning approaches.** (A) Conventional convolutional neural network (CNN) that trains a model using a single-pass or one-step transfer approach, and (B) proposed cumulative transfer learning CNN, where knowledge is progressively integrated across multiple phases.

introduced to the model incrementally, allowing the network to acquire new insights while preserving earlier learned parameters. Between each incremental phase, we performance the evaluations *via* metrics such as accuracy, precision, or recall to ensure the newly assimilated data enhances rather than overwrites existing knowledge (*i.e.*, mitigating the issue of "catastrophic forgetting"). Over successive phases, this adaptive inheritance of learned representations yielded robust performance, improved generalization, and accelerated convergence relative to conventional single-stage training. Such a tiered approach is especially advantageous for domains afflicted by limited or imbalanced datasets, as it maximizes the utility of previously gleaned insights, fostering more nuanced feature extraction and heightened classification accuracy in dynamic or evolving tasks (*Zorić et al., 2024*). By incorporating knowledge transfer at multiple stages, this method allows the model to capture nuanced morphological features of *E. vermicularis* eggs, including elliptical structure, shell thickness, and internal granularity. These subtle characteristics are often overlooked by traditional methods (*Xu et al., 2016*).

Cumulative transfer learning, step-by-step:

Assume we have three datasets

$$Dataset\ A:\ D_A = \left\{ \left( x_A^i, y_A^i \right) \right\}_{i=1}^{N_A}$$
$$Dataset\ B:\ D_B = \left\{ \left( x_B^i, y_B^i \right) \right\}_{i=1}^{N_B} \quad (1)$$
$$Dataset\ C:\ D_C = \left\{ \left( x_C^i, y_C^i \right) \right\}_{i=1}^{N_C}$$

where $x^i$ represents an input sample, $y^i$ is the corresponding label, $N$ is the number of samples in each dataset.

Step 1: Pre-training on Dataset A

The first model is trained to minimize a loss function $\zeta_A$

$$\theta_A^* = argmin\ \mathrm{E}_{(x,y)\sim P_A}[\zeta_A(f(x;\theta),y)] \tag{2}$$

Using gradient descent algorithm (Eq. (3)):

$$\theta_A^* = \theta_A - \eta\frac{\partial\zeta_A}{\partial\theta} \tag{3}$$

where $\theta_A^*$ represent the optimal parameters for dataset A, $\eta$ is learning rate, $\frac{\partial\zeta_A}{\partial\theta}$ is the gradient loss function.

Step 2: Fine-tuning on Dataset B

We initialize the model for Dataset B with $\theta_A^*$, instead of random initialization

$$\theta_B^* = argmin\ \mathrm{E}_{(x,y)\sim P_B}[\zeta_B(f(x;\theta),y)]. \tag{4}$$

Using gradient descent algorithm on 2$^{nd}$ transfer learning iteration (Eq. (5)):

$$\theta_B^* = \theta_A^* - \eta\frac{\partial\zeta_B}{\partial\theta}. \tag{5}$$

Step 3: Fine-tuning on Dataset C

We initialize the model for Dataset B with $\theta_A^*$, instead of random initialization

$$\theta_C^* = argmin\ \mathrm{E}_{(x,y)\sim P_C}[\zeta_C(f(x;\theta),y)]. \tag{6}$$

Using gradient descent algorithm on 3$^{rd}$ transfer learning iteration (Eq. (7)):

$$\theta_C^* = \theta_B^* - \eta\frac{\partial\zeta_C}{\partial\theta}. \tag{7}$$

The algorithms, codes, and README file are available at the following GitHub repository: https://github.com/Pongphan/ctransferlearning_ev (accessed on 22 March 2025).

## Evaluation metrics

A comprehensive set of evaluation metrics was employed to assess model performance. Accuracy measured the proportion of correctly classified images relative to the total number of samples. Precision quantified the proportion of positive identifications that were truly positive, indicating the reliability of positive predictions. Recall measured the model's ability to identify all relevant instances, ensuring minimal misses true positives (TPs). The F1-score, as the harmonic mean of precision and recall, provided a balanced performance measure particularly useful for class imbalance scenarios. Sensitivity and specificity were included for their clinical significance. Sensitivity measured the ability to detect TPs, while specificity evaluated the capacity to correctly identify true negatives. These metrics provided a multi-dimensional understanding of model performance, with particular emphasis on minimizing false negatives (FNs) due to their critical importance in clinical diagnostics.

## Comparison to baseline models

To assess the efficacy and robustness of our proposed cumulative transfer learning framework, we conducted a comprehensive comparative analysis against several widely recognized deep neural network architectures that commonly employ in medical image classifications. Specifically, we selected conventional CNN (*Chaibutr et al., 2024*), ResNet50, InceptionV3, DenseNet121, and Xception as benchmark models due to their established track records in handling complex visual patterns, their proven capability to achieve high accuracy on large-scale image recognition tasks, and their broad adoption within the medical imaging community (*Gunturu et al., 2024*). Conventional CNN, the architecture begins with a sequence of convolutional layers, each utilizing 64 filters with a 3 × 3 kernel size. After every two convolutional layers, max pooling is applied to downsample the spatial dimensions of the feature maps, thereby reducing computational complexity and emphasizing the most critical features. To further prevent overfitting, dropout is introduced following the max-pooling layers, randomly omitting units during training. The model then progresses from convolutional operations to fully connected layers, comprising three dense layers with 128, 64, and eight units, respectively. Each dense layer is also followed by batch normalization and dropout to maintain regularization. The final dense layer contains two units, making it appropriate for binary classification tasks. This architecture, characterized by its depth and the integration of multiple regularization techniques, is designed to effectively manage complex data while minimizing the risk of overfitting (*Chaibutr et al., 2024*). ResNet50, with its residual connections, mitigates the issues related to the vanishing gradients and has been shown to excel in extracting detailed hierarchical features (*Rahmati, Shirani & Keshavarz-Motamed, 2024*). InceptionV3's inception modules promote an efficient parallel feature extraction across multiple scales, while DenseNet121's densely connected layers facilitate feature reuse, ultimately reducing model parameters without compromising performance (*Hu et al., 2021*). Similarly, Xception's depthwise separable convolutions are enable a more efficient factorization of convolutional operations, potentially improving both accuracy and computational efficiency (*Peng & Wang, 2021*). By comparing our approach against these influential and architecturally distinct networks, we aimed to establish a robust benchmark and discerned the relative advantages were conferred by our cumulative transfer learning strategy in enhancing the sensitivity, specificity, and overall diagnostic reliability of *E. vermicularis* egg detection. To compare the performance of object detection models, IoU is commonly used as a standard metric. IoU quantifies the spatial overlap between a predicted bounding box and the corresponding ground truth annotation, calculated as the area of intersection divided by the area of union between the two boxes. This provides an objective measure of localization accuracy.

For comparative analysis, we also implemented the You Only Look Once version 8 (YOLOv8) object detection framework as a baseline. YOLOv8 is a state-of-the-art, single-stage object detection model renowned for its speed and accuracy in diverse computer vision tasks (*Ko & Lee, 2025*). The model was trained on the same annotated dataset comprising images of *E. vermicularis* eggs and artifacts, following identical

preprocessing and augmentation protocols as used for the cumulative transfer learning approach. Training parameters including input image size, learning rate, and batch size were optimized to ensure fair comparison. Detection performance was quantitatively evaluated using IoU metrics, with predicted bounding boxes compared against expert-annotated ground truth on the test set. This benchmarking enabled direct assessment of the relative strengths and limitations of the YOLOv8 model in the context of automated parasitological diagnostics.

## RESULTS

### Dataset

All images used for training were standardized by resizing them to $128 \times 128$ pixels to ensure uniformity across the dataset. Initially, 1,000 images were collected for each of the two classes namely, artifacts (class 0) and *E. vermicularis* eggs (class 1) resulting in a balanced starting dataset of 2,000 images. To enhance the diversity and robustness of the training data, a multi-step augmentation procedure was implemented, beginning with random rotations followed by sequential application of a mean filter, Gaussian blur, Gaussian noise, and sharpening. Each of these transformations was systematically applied to increase the dataset by a factor of four for each original image, ultimately yielding 4,000 augmented images per class and contributing to improved model generalization. For evaluation, an independent testing set comprising 100 images of artifacts (class 0) and 100 images of *E. vermicularis* eggs (class 1) was curated under the same resolution constraints. This clearly separated test set enabled an unbiased assessment of the trained model's performance on previously unseen data.

### Conventional learning

After completing the five-fold cross-validation procedure, in which the dataset is partitioned into five equally sized subsets such that each subset serves as the validation set exactly once, Fig. 3A showed line plots of classification accuracy for each fold. Error bars were included to illustrate variability across partitions, thereby demonstrating the model's performance consistency under different data splits. In the same Fig. 3B, the receiver operating characteristic (ROC) curve illustrated the trade-off between sensitivity (TP rate) and 1—specificity (FP rate), with a diagonal reference line indicating a random classification. The area under the ROC curve (ROC-AUC) quantified the model's discriminative power, where values approaching 1.0 signify high diagnostic efficacy. Taken together, the fold-specific accuracy trends and ROC analysis provided a comprehensive assessment of the model's robustness and predictive capability under varying training validation configurations.

### Cumulative transfer learning

In our cumulative transfer learning paradigm, the training process was subdivided into five consecutive rounds, each determined by the augmented data generated at each stage. Within each round, we applied a five-fold cross-validation, partitioning the dataset into five unique subsets to evaluate model performance and enhance generalization ability.

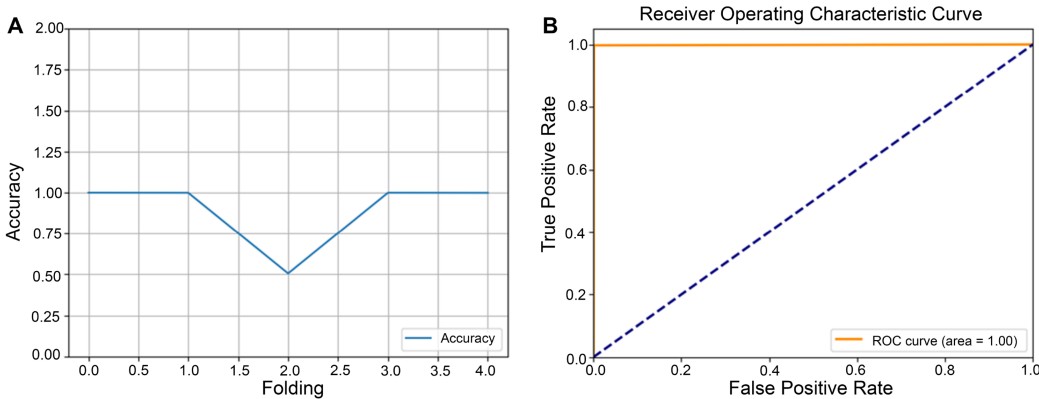

**Figure 3 Performance evaluation of the model using five-fold cross-validation.** The dataset was partitioned into five equally sized subsets following a five-fold cross-validation scheme. Each subset served as the validation set in turn, while the remaining subsets were used for model training. (A) Line plots depict the classification accuracy across each fold. (B) Receiver operating characteristic (ROC) curve illustrating the model's classification performance.

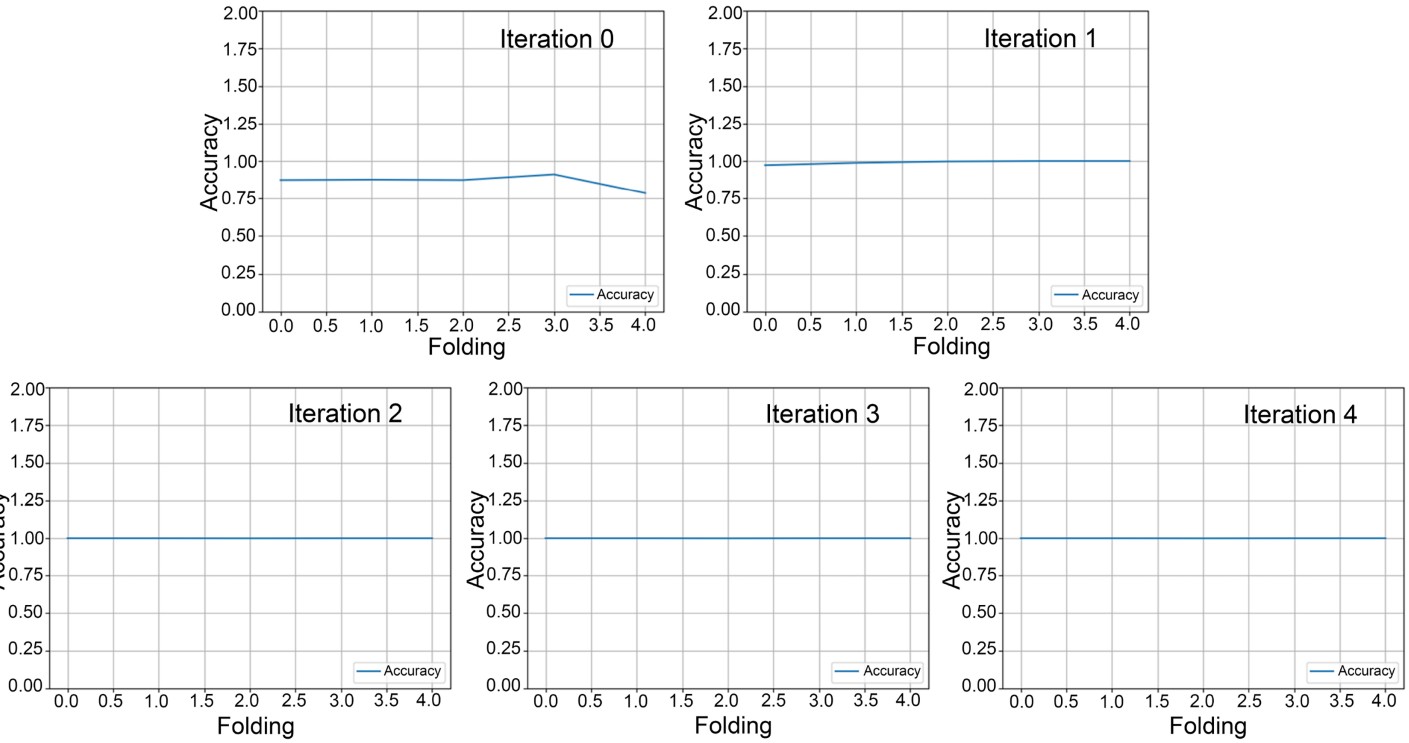

**Figure 4 Five-fold cross-validation accuracy across rounds.** Five-fold cross-validation results from round 0 to round 4, showing model accuracy for each fold. Each line represents the classification accuracy of a specific fold, demonstrating the model's consistency and generalizability across different data subsets.

Figures 4 and 5 presented the outcomes of this iterative five-fold cross-validation from round 0 to round 5, encompassing the entire progression of the cumulative transfer learning approach. Each point reflects the model's average performance across the five

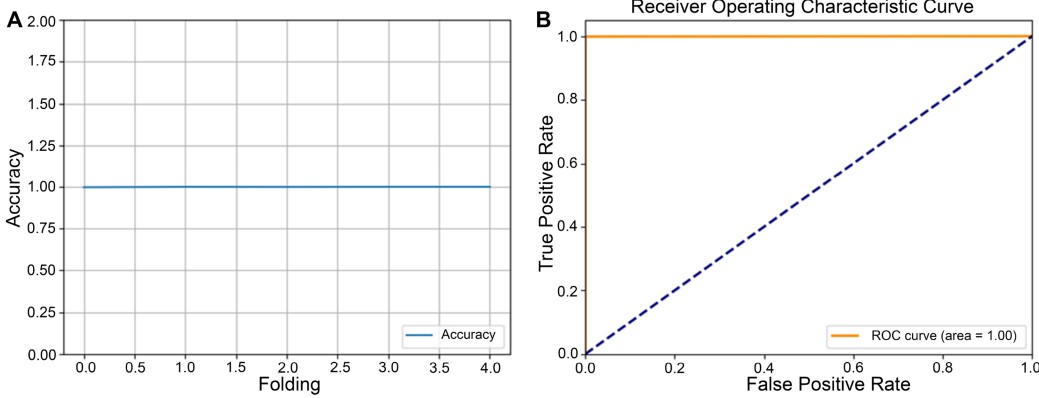

**Figure 5  Five-fold cross-validation accuracy and receiver operating characteristic (ROC) curve for round 5.** (A) Five-fold cross-validation accuracy for round 5, illustrating each fold's classification performance and providing an overview of the model's consistency across the dataset. The accuracy per fold highlights the system's reliability and potential for generalization. (B) ROC curve for round 5, demonstrating the model's performance in distinguishing between classes.

validation folds, providing robust estimates of classification accuracy, loss, and other relevant metrics. The figures are organized into three key sections: (1) cross-validation loss for each fold, (2) accuracy per fold, and (3) the ROC curves. This framework illustrates the model's learning trajectory in each round and demonstrates improvements in both classification performance and generalizability. The cross-validation loss plots highlight how the model's error evolved across progressive rounds. Lower loss values in later rounds indicated enhanced ability to detect *E. vermicularis* eggs, reflecting successful knowledge transfer in cumulative learning. The middle section presents the accuracy achieved in each of the five folds, showing how classification performance improves from round 0 to round 5. Rising accuracy from one round to the next underscores the positive impact of incremental transfer learning on the feature extraction and classification tasks. The rightmost section in Fig. 5 shows ROC curves, providing an in-depth view of the TP rate against the FP rate for each fold. By comparing curve shapes and their corresponding area under the curve (AUC) values, researchers can evaluate both the sensitivity and specificity gains throughout the cumulative transfer learning process. Collectively, these panels offer comprehensive performance evaluation. Figure 5 underscores the robustness of the proposed cumulative transfer learning approach. Moreover, the five-fold cross-validation design to ensure that the reported performance is not overly dependent on any particular data partition, thereby reinforcing the reliability of these findings for clinical applications.

As illustrated in Table 2, a detailed comparison between conventional CNN and cumulative transfer learning revealed a modest, yet appreciable improvement in model performance, as evidenced by a slight increase in overall accuracy when employing the cumulative transfer strategy. Beyond the accuracy gain, a significant reduction in training time was observed: the cumulative transfer learning model required only 0:51:44 h for completion, compared to the 2:07:32 h needed by the conventional approach. This acceleration largely contributed to more efficient parameter initialization and the

**Table 2 Comparison of training time, accuracy, and RAM usage between conventional convolutional neural network (CNN) and the proposed cumulative transfer learning CNN approach.**

| Learning method | Iteration | Time (h) | Accuracy | RAM usage (GB) |
|---|---|---|---|---|
| Conventional CNN | None | 2:07:32 | 0.94 | 56.7 |
| Proposed cumulative transfer learning CNN | 0 | 0:14:09 | 0.57 | |
| | 1 | 0:34:03 | 0.95 | |
| | 2 | 0:00:34 | 0.95 | |
| | 3 | 0:00:50 | 0.96 | |
| | 4 | 0:01:02 | 0.95 | |
| | 5 | 0:01:03 | 0.87 | 19.4 |

**Table 3 Performance metrics (accuracy, precision, recall, F1-score) of various deep learning models for *Enterobius vermicularis* egg classification.**

| Model | Accuracy | Precision | Recall | F1-score |
|---|---|---|---|---|
| Conventional convolutional neural network (CNN) | 0.94 | 0.94 | 0.94 | 0.94 |
| Proposed cumulative transfer learning CNN | 1.00 | 1.00 | 1.00 | 1.00 |
| ResNet50 | 0.99 | 0.99 | 0.99 | 0.98 |
| InceptionV3 | 1.00 | 1.00 | 1.00 | 1.00 |
| DenseNet121 | 0.96 | 0.96 | 0.96 | 0.96 |
| Xception | 0.97 | 0.97 | 0.97 | 0.97 |

progressive reuse of knowledge across training phases, thereby minimizing redundancy in feature extraction.

In addition to these time savings, the cumulative transfer learning also demonstrated a marked decrease in resource consumption, where the peak memory usage dropped from 56.7 to 19.4 GB. Such a substantial reduction was not only lowers computational costs but can also expanded the feasibility of deploying advanced DL models in resource-constrained settings. Collectively, these findings underscored the potential value of cumulative transfer learning in both enhancing diagnostic accuracy and optimizing computational efficiency for *E. vermicularis* egg detection.

In this study, we conducted a comparative evaluation of four DL architectures ResNet50, InceptionV3, DenseNet121, and Xception as summarized in Table 3. The goal was to assess each model's performance in classifying *E. vermicularis* eggs acquired from scotch tape samples. Of the tested architectures, two models emerged with superior metrics: our proposed the cumulative transfer learning CNN and the InceptionV3 network. Notably, both systems achieved a perfect classification performance, demonstrated by an accuracy, precision, recall, and F1-score of 1.0. These results underscored the effectiveness of cumulative transfer learning, which enabled iterative refinement across multiple training phases, thereby leveraging knowledge from prior training steps to enhance final predictive capabilities. Furthermore, the notable success of InceptionV3 recognized for its multi-branch convolution modules and dimensional reduction techniques reinforced that the advanced DL methodologies can outperform

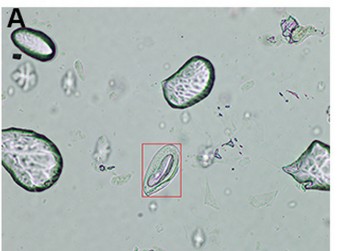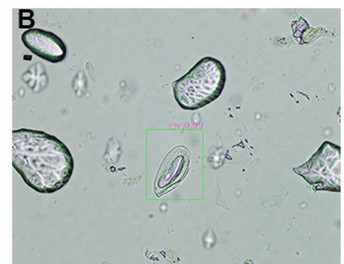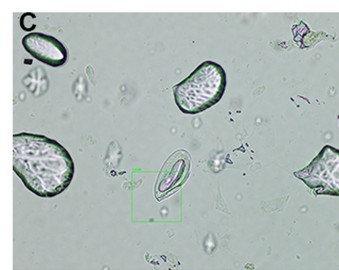

**Figure 6 Comparison of object detection outcomes: expert annotation *vs* model predictions.** A detailed comparison of object detection results in microscopic images, highlighting differences between expert medical annotations and machine learning model predictions. (A) Red bounding box indicates objects identified by expert personnel. (B) Green bounding box represents annotations generated by the proposed cumulative transfer learning convolutional neural network (CNN) model. (C) Green bounding box represents annotations generated by the you only look once version 8 (YOLOv8) model.

traditional baselines in detecting parasitic eggs. Achieving perfect evaluation metrics in this diagnostic setting highlights the promise of these approaches for reliable, high-throughput laboratory implementation.

## Object detection

A comprehensive comparative analysis of object detection outputs generated by the high-precision ML models, compared against expert medical annotations on microscopic images, is presented in Fig. 6. By juxtaposing these two sets of annotations, one can directly evaluated the model's detection accuracy, including its ability to localize and identify target structures, as well as to identify any discrepancies that may inform further refinement of the detection algorithm. In Fig. 6A, red bounding box represented the expert-verified ground truth, enabling clear visualization of clinically relevant target. The automatically generated bounding boxes are highlighted in green from the proposed model (Fig. 6B) and YOLOv8 model (Fig. 6C).

An in-depth overview of the IoU thresholds obtained by the proposed model was displayed in Table 4. Comparative analysis of the proposed cumulative transfer learning CNN and YOLOv8 reveals distinct performance characteristics. The proposed cumulative transfer learning approach achieved perfect sensitivity (TP = 6, FN = 0) but exhibited a high FP rate (FP = 97). Its average precision (AP) @IoU0.5 (0.530) substantially outperformed YOLOv8 (0.245), indicating superior detection precision at the standard IoU threshold. However, its mean average precision (mAP) @IoU0.5:0.05:0.95 (0.027) was considerably lower than YOLOv8 (0.057), suggesting limited generalization across stricter IoU thresholds. In contrast, YOLOv8 demonstrated higher detection capability (TP = 23) with fewer FPs (FP = 71) but suffered from missed detections (FN = 9) and lower precision at IoU0.5. The superior mAP of YOLOv8 reflects greater robustness across varying overlap criteria, likely attributable to its anchor-free architecture and optimized localization-classification balance. These results indicate that the proposed cumulative transfer learning model prioritizes sensitivity and single-threshold precision, while

**Table 4 Overview of the intersection over union (IoU) threshold ranges achieved by the proposed cumulative transfer learning convolutional neural network (CNN) model and the you only look once version 8 (YOLOv8) model.**

| Object detection model | True positive | False positive | False negative | Average precision (AP) @IoU0.5 | Mean average precision (mAP) @IoU0.5:0.05:0.95 |
|---|---|---|---|---|---|
| Proposed cumulative transfer learning CNN | 6 | 97 | 0 | 0.530 | 0.027 |
| YOLOv8 | 23 | 71 | 9 | 0.245 | 0.057 |

YOLOv8 demonstrates superior overall localization stability. The findings suggest that model selection should be guided by application-specific requirements regarding tolerance for FNs *vs* localization accuracy demands.

## DISCUSSION

The scotch tape technique has been widely used as the gold standard for diagnosing *E. vermicularis* infections due to its simplicity and effectiveness in capturing eggs from the perianal region (*Garcia & Procop, 2016*). This method provides a non-invasive and cost-effective diagnostic approach, making it particularly valuable in resource-limited settings and for large-scale epidemiological studies (*Quistberg et al., 2024*). However, despite its widespread use, the technique has the drawbacks. The scotch tape often collects a variety of artifacts, including debris, epithelial cells, and other non-parasitic structures, which can obscure the microscopic visualization of *E. vermicularis* eggs (*Reinhard, Araújo & Morrow, 2016*). These artifacts introduce considerable variability and complexity to the diagnostic process, increasing the reliance on specialized expertise for accurate identification.

The subjective nature of manual microscopy further compounds this issue, as diagnostic accuracy may vary depending on the skill and experience of the laboratory personnel. In addition, the time-intensive nature of manual examination makes it impractical for handling large volumes of samples in high-burden areas. These challenges underscore the need for more automated and objective diagnostic methods to supplement or replace traditional approaches, particularly in settings where trained personnel are scarce. Emerging computational tools, such as ML-based object detection systems, hold significant promise in overcoming these limitations by providing rapid, reproducible, and accurate detection of *E. vermicularis* eggs (*Litjens et al., 2017*). Such innovations could greatly enhance diagnostic workflows, reducing both human error and time-to-result, while maintaining or even improving diagnostic precision.

Deep learning paradigms have consistently demonstrated significant potential in improving diagnostic precision, efficiency, and reproducibility in medical image analysis (*Litjens et al., 2017*). These advancements stem from their ability to extract complex patterns and features from data, surpassing traditional methods that rely on handcrafted features or shallow classifiers. Cumulative transfer learning extends this capability by iteratively refining learned representations across multiple training stages. This approach leverages prior knowledge to improve performance in data-scarce scenarios, which is particularly advantageous in parasitological research where labeled datasets are often

limited (*Pan & Yang, 2009*; *Wang & Deng, 2018*; *Raghu & Schmidt, 2020*). By incorporating knowledge transfer at multiple stages, this method allows the model to capture the nuanced morphological features of *E. vermicularis* eggs. These features include elliptical structure, shell thickness, and internal granularity. Such subtle characteristics might be overlooked by traditional or less sophisticated methods (*Xu et al., 2016*).

Cumulative transfer learning is particularly advantageous in domains where dataset heterogeneity is constrained, yet achieving high diagnostic specificity is critical, such as in clinical diagnostics and large-scale epidemiological monitoring. A notable gap in the current literature is the absence of cumulative transfer learning frameworks specifically designed for automated detection of *E. vermicularis* eggs, particularly in the context of artifact-rich and data-scarce parasitological images (*Liang et al., 2016*; *Holmström et al., 2017*; *Pho et al., 2022*; *Penpong et al., 2023*; *Wan et al., 2023*; *Chaibutr et al., 2024*; *He et al., 2024*; *Lundin et al., 2024*; *Thanchomnang et al., 2024*; *Xu et al., 2024*). In this study, we introduce a novel framework designed for the automated detection of *E. vermicularis* eggs, marking its first documented application in parasitological image analysis. Unlike conventional transfer learning approaches that apply static feature reuse from pre-trained convolutional networks, our method employs progressive model adaptation across multiple training distributions. By leveraging iterative fine-tuning and domain-specific feature distillation, our approach facilitates incremental parameter optimization, enabling a more effective representation shift toward task-relevant morphological features. This cumulative refinement process mitigates the risk of catastrophic forgetting while ensuring improved feature generalization across microscopic imaging variations. Compared to standard transfer learning strategies that rely on one-time adaptation from a base model (*Chaibutr et al., 2024*), our multi-stage optimization paradigm demonstrates superior performance. The empirical improvements highlight the efficacy of structured parameter evolution, reinforcing the utility of cumulative transfer learning as a robust alternative for DL applications in specialized medical image classification tasks.

By addressing the inherent challenges in parasitological diagnostics, our proposed algorithm validates the efficacy of cumulative transfer learning in achieving high diagnostic robustness and reproducibility, establishing a strong foundation for broader deployment in related biomedical imaging tasks. The results emphasize the scalability of our approach, demonstrating its potential to enhance routine diagnostic pipelines and enable efficient, high-precision detection frameworks in resource-constrained environments. To mitigate the risk of overfitting and ensure model generalization, we adopted a dual-pronged evaluation strategy integrating stratified validation splitting with five-fold cross-validation. This methodological framework ensured a statistically rigorous assessment, minimizing estimation bias while controlling variance across training subsets (*Ying, 2019*). The experimental analysis was conducted on a dataset comprising 103 high-resolution microscopic images, each containing one instance of *E. vermicularis* eggs. This simulated real-world diagnostic conditions in clinical and laboratory settings. The progressive knowledge refinement in cumulative transfer learning enabled the model to systematically learn domain-invariant yet task-specific morphological features. This approach reinforced adaptability to variations in staining, magnification, and sample

preparation artifacts, which are key challenges in automated parasitology detection. Model performance was quantitatively assessed using the IoU metric, a widely recognized standard in object detection. This metric measures spatial congruence between predicted bounding boxes and ground truth annotations. Instances with moderate IoU values were primarily characterized by model-generated bounding boxes exceeding the expert annotations in size. This tendency suggests an implicit regularization effect, where the network adopts a conservative detection strategy by expanding the bounding box perimeter. Such a bias toward larger bounding boxes may serve as an adaptive mechanism to mitigate the risk of under-detection, particularly in cases where egg boundaries exhibit high inter-sample variability or are partially occluded. While this phenomenon may marginally affect precision scores, it aligns with a risk-averse diagnostic paradigm that prioritizes sensitivity in clinical applications, reducing the likelihood of FNs, which is a critical consideration in automated parasite detection workflows (*Chaibutr et al., 2024*). The implementation of YOLOv8 as a benchmark revealed its ability to detect most test images; however, these results demonstrate a trade-off between the two models. The proposed cumulative transfer learning CNN achieves perfect sensitivity and high precision at a single IoU0.5, but its high FP rate and low mAP@IoU0.5:0.05:0.95 indicate poor localization consistency across thresholds. Conversely, YOLOv8 provides better generalization and fewer FPs but lower single-threshold precision. Thus, model selection should depend on whether the application demands maximum sensitivity or broader detection robustness. When implementing automated detection in clinical applications, laboratory personnel must remain vigilant regarding potential FP and FN results. To mitigate diagnostic errors, we recommend a two-tier verification approach: (1) manual verification of all positive detections through examination of bounding boxes on digitized images to confirm the presence of actual *E. vermicularis* eggs before reporting results, and (2) systematic microscopic re-examination of negative cases for final confirmation. This dual-verification protocol ensures clinical accuracy while maintaining the efficiency gains of automated detection systems. Furthermore, real-time detection remains limited due to the extended computational time required by the cumulative transfer learning algorithm.

## CONCLUSIONS

This study highlights the potential of an innovative cumulative transfer learning algorithm to enhance parasitological diagnostics by enabling accurate, scalable, and automated detection of *E. vermicularis* eggs, even in artifact-laden scotch tape preparations. By leveraging iterative model adaptation and domain-specific feature distillation, the proposed framework mitigates catastrophic forgetting, improves feature generalization, and outperforms conventional transfer learning methods. Empirical results validate its robustness and scalability, particularly in resource-constrained settings, reinforcing its applicability for routine clinical workflows and large-scale public health surveillance. The proposed cumulative transfer learning approach achieved perfect sensitivity in detecting all *E. vermicularis* eggs. However, the relatively low mAP indicates challenges in precise localization and boundary delineation. While the model excels at identifying target objects, bounding box precision requires further optimization. Further research is needed to

expand dataset diversity, refine IoU performance, and integrate explainable AI frameworks to enhance transparency and clinician trust.

## ACKNOWLEDGEMENTS

We extend our gratitude to the anonymous reviewers for their insightful feedback and constructive suggestions that substantially enhanced the quality of this article.

### Funding

This project was funded by the grant from the National Research Council of Thailand (NRCT): High-Potential Research Team Grant Program (Contract no. N42A670561 to Wanchai Maleewong (WM)). The contents of this report are solely the responsibility of the authors and do not necessarily represent the official views of the NRCT. This work has been supported by Walailak University Graduate Scholarships (Contract no. 28/2023). This research work is financially supported by Walailak University Graduate Research Fund (Contract no. CGS-RF-2025/03). The funders had no role in study design, data collection and analysis, decision to publish, or preparation of the manuscript.

### Grant Disclosures

The following grant information was disclosed by the authors:
National Research Council of Thailand (NRCT).
High-Potential Research Team Grant Program: N42A670561.
Walailak University Graduate Scholarships: 28/2023.
Walailak University Graduate Research Fund: CGS-RF-2025/03.

### Competing Interests

The authors declare that they have no competing interests.

### Author Contributions

- Pongphan Pongpanitanont conceived and designed the experiments, performed the experiments, analyzed the data, performed the computation work, prepared figures and/or tables, authored or reviewed drafts of the article, and approved the final draft.
- Naparat Suttidate conceived and designed the experiments, analyzed the data, authored or reviewed drafts of the article, and approved the final draft.
- Hiroshi Yamasaki performed the experiments, authored or reviewed drafts of the article, and approved the final draft.
- Wanchai Maleewong conceived and designed the experiments, authored or reviewed drafts of the article, and approved the final draft.
- Penchom Janwan conceived and designed the experiments, performed the experiments, analyzed the data, performed the computation work, prepared figures and/or tables, authored or reviewed drafts of the article, and approved the final draft.

## Ethics

The following information was supplied relating to ethical approvals (*i.e.*, approving body and any reference numbers):

This study was conducted in accordance with the Declaration of Helsinki and approved a certificate of exemption by the Ethics Committee in Human Research Walailak University (protocol code WUEC-24-387-01 and date of approval 8 November 2024).

## Data Availability

The algorithms, codes, and README file are available at GitHub and Zenodo:

- https://github.com/Pongphan/ctransferlearning_ev.

- Pongphan Pongpanitanont. (2025). Pongphan/ctransferlearning_ev: ctransferlearning (ctransferlearning). Zenodo. https://doi.org/10.5281/zenodo.16943242.

This study used *E. vermicularis* egg images available at Figshare: Chaibutr, Natthanai; Pongpanitanont, Pongphan; Laymanivong, Sakhone; Thanchomnang, Tongjit; Janwan, Penchom (2024). Figure. figshare. Dataset. https://doi.org/10.6084/m9.figshare.26266028.v2.

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
