# Peer review of "Enhanced object detection of *Enterobius vermicularis* eggs using cumulative transfer learning algorithm"

_PeerJ Computer Science, doi:10.7717/peerj-cs.3213_

## Round 0.1 · original submission · Major Revisions

Reviewer 1 ·

Basic reporting

Strengths:
• The manuscript is well-written in clear and professional English, making it accessible to an international audience.
• The introduction provides a solid context with relevant background information and adequate references to current literature.
• The structure of the manuscript adheres to the standards of PeerJ and the norms of the relevant scientific discipline, ensuring a well-organized flow of information.
Areas for Improvement:
• While the introduction is generally strong, it could benefit from a more detailed explanation of the specific knowledge gap that this study aims to address.
• While the abstract need more detailed explanation in the methods content.
• Some figures and tables lack detailed metadata or captions, which may require additional effort from readers to fully understand the visual content.

Experimental design

Strengths:
• The research falls within the scope of the journal and aligns with the scientific objectives of the fields of AI and Image Processing.
• The experimental methods are described in detail, including the use of high-resolution datasets, computational platforms (Google Colab with NVIDIA A100 GPU), and data augmentation procedures such as Gaussian blur, noise addition, and sharpening to enhance model generalization.
• The validation protocol employs five-fold cross-validation appropriately, ensuring a more reliable evaluation of model performance.
• Technical aspects such as data preprocessing are discussed comprehensively, including augmentation techniques to handle variations in real clinical image quality.
Areas for Improvement:
• The authors could provide more detailed information about the source code or scripts used in the study to facilitate replication by other researchers (e.g., a link to a GitHub repository).
• Although ethical considerations are mentioned, they could be strengthened, particularly regarding the management of sensitive data if applicable.

Validity of the findings

Strengths:
• The experimental results are supported by rigorous statistical evaluations through stratified validation splitting and bootstrap aggregation (bagging) to reduce bias in estimates and enhance the stability of performance metrics.
• The conclusions are drawn from empirical results without overclaiming; the authors also acknowledge the limitations of the study and suggest directions for future research as a critical reflection on their findings.
Areas for Improvement:
• The error analysis could be further elaborated, especially concerning false positive/negative cases, to provide readers with a clearer understanding of the risks associated with model detection in real clinical applications.

Additional comments

Overall, the manuscript demonstrates high quality in terms of basic reporting, experimental design, and validity of findings. However, some minor improvements are needed to enhance transparency and replicability of the results.

I recommend a Minor Revision, focusing on improving technical documentation (code/resources), elaborating on the motivation in the introduction, and deepening the analysis of model detection errors.

Annotated reviews are not available for download in order to protect the identity of reviewers who chose to remain anonymous.

Reviewer 2 ·

Basic reporting

1. The abstract does not mention the dataset size, source, or if it was publicly available. Briefly state the number of images used, or whether it was a private/public dataset, to help assess reproducibility.
2. Clarity of terminology, grammar, and presentation of evaluation results in the abstract can be improved.
3. Some sentences are overly long and contain multiple clauses, which can make them harder to follow. Split the long sentences into multiple shorter sentences for clarity.
4. The authors mention addressing limitations of previous studies, but the discussion could be improved by linking the specific gaps and limitations identified in Table 1 with the proposed improvements.
5. The motivation behind using the scotch tape technique and the dataset is clear. However, the rationale behind certain methodological choices, such as the specific augmentation techniques and model architecture, could be better justified in terms of their expected benefits for egg detection.
6. Introduction section could contain example images of different types of images from the dataset indicating various artifacts and eggs.

Experimental design

1. Comparing the results of the proposed methodology with recent studies on detecting similar types of parasite eggs could further strengthen this study.
2. Explicitly connect the proposed method to the gaps identified in previous research and provide more detailed references on similar techniques and datasets used in the field.
3. In object detection, mean Average Precision (mAP) is often preferred—its absence is a missed opportunity.
4. No quantitative comparison with other detection models (e.g., YOLO, RetinaNet, standard transfer learning). Including baseline results would further highlight the efficacy of the CTL approach.

Validity of the findings

No comments

Additional comments

1. Automating parasitological diagnosis (specifically Enterobius vermicularis detection) is highly relevant, especially for remote or resource-limited settings and the authors clearly explain why traditional diagnostic methods are insufficient.
2. The use of cumulative transfer learning appears to be a novel approach for this application however addition of a short clarification of the same may help readers understand the innovation.
3. The conclusion clearly encapsulates the core findings without redundancy. However, including a brief mention of the achieved IoU score or accuracy could have added weight to the claims.
4. Practical issues like computational requirements or potential barriers to real-time use are not mentioned, which could matter for clinical deployment.

Reviewer 3 ·

Basic reporting

1. Review the figures. They are illegible.
2. Review the journal template.

Experimental design

1. I disagree with the choice of representation for Figure 3. The accuracy does not exceed 1, whereas the author represents it up to scale 2. In addition, iterations 1, 2, 3, and 4 are at 100%. I think there is overfitting.
2. Figure 4 also shows that the model learns quickly. It confirms that the dataset is too small to be used as a learning base for the proposed model. It is necessary to review the size of the elements of the base or the size of the base itself

Validity of the findings

no comment

Additional comments

no comment

---

## Round 0.2 · accepted · Accept

The authors have satisfied the reviewers' requests, and so I can recommend this article for acceptance

Reviewer 2 ·

Basic reporting

All comments of the reviewer are addressed appropriately.

Experimental design

All suggestions are incorporated in the revised manuscript.

Validity of the findings

-

Additional comments

The manuscript has been revised as per the suggestions.